# A ventral hippocampal-lateral septum pathway regulates social novelty preference

**Maha Rashid[1], Sarah Thomas[2], Jennifer Isaac[1], Sonia Corbett Karkare[2], Hannah Klein[2], Malavika Murugan[2,3]\***

[1]Emory Neuroscience Graduate Program, Emory University, Atlanta, United States; [2]Department of Biology, Emory University, Atlanta, United States; [3]Center for Translational Social Neuroscience, Emory University, Atlanta, United States

## eLife assessment

This **important** manuscript uses circuit mapping, chemogenetics, and optogenetics to demonstrate a novel hippocampal lateral septal circuit that regulates social novelty behaviours and shows that downstream of the hippocampal septal circuit, septal projections to the ventral tegmental area are necessary for general novelty discrimination. The strength of the evidence supporting the claims is **convincing** but would be strengthened by the inclusion of additional functional assays. The work will be of interest to systems and behavioural neuroscientists who are interested in the brain mechanisms of social behaviours.

**\*For correspondence:**
mmurug5@emory.edu

**Abstract** The ability to distinguish strangers from familiar individuals is crucial for the survival of most mammalian species. In humans, an inability to recognize kin and familiar individuals and engage in appropriate behaviors is associated with several types of dementia, including Alzheimer's disease. Mice preferentially spend more time investigating a novel individual relative to a familiar individual. Yet, how social novelty-related information drives increased investigation of the novel animal remains poorly understood. Recent evidence has implicated the ventral hippocampus (vHPC) as a key node in encoding information about conspecific identity. Of particular interest are vHPC projections to the lateral septum (LS), a region that has been implicated in driving a wide range of motivated social behaviors. In this study using chemogenetics, optogenetics, and monosynaptic rabies tracing, we identified a novel vHPC-LS-ventral tegmental area (VTA) pathway that is necessary for mice to preferentially investigate novel conspecifics. Using monosynaptic rabies tracing, we established that LS neurons make direct monosynaptic connections onto dopaminergic neurons in the VTA. Thus, we have identified a potential pathway via which conspecific identity could be transformed to drive motivated social behaviors.

## Introduction

Most animals make daily calculations to approach or avoid a conspecific based on the quality of prior interactions. Yet, how we recognize who we interact with and how that information is transformed to guide subsequent behavior remains poorly understood. For instance, mice readily discriminate between novel and familiar conspecifics, preferentially spending more time investigating a novel individual relative to a familiar individual (*Moy et al., 2004*). However, it remains unknown how social novelty drives increased approach and investigation.

The vHPC and dorsal CA2 neurons are known to be causally involved in allowing mice to distinguish between novel and familiar conspecifics (*Hitti and Siegelbaum, 2014*; *Okuyama et al., 2016*; *Phillips et al., 2019*; *Donegan et al., 2020*; *Tao et al., 2022*). Specifically, studies have shown that manipulating the activity of ventral CA1 but not dorsal CA1 neurons in the hippocampus disrupts the ability of mice to discriminate between novel and familiar conspecifics (*Okuyama et al., 2016*). Notably, individual vHPC neurons are known to preferentially fire in the proximity of conspecifics but not objects (*Rao et al., 2019*). Additionally, experiments have shown that ensembles of neurons in the vHPC encode the identity of a conspecific and that this representation changes as a function of familiarity with the conspecific (*Okuyama et al., 2016*). However, it is unclear how this social memory information in the vHPC is transformed to preferentially drive increased investigation of a novel conspecific. Moreover, there remains debate regarding which regions downstream of the vHPC play a role in promoting differential investigation of novel and familiar conspecifics (*Okuyama et al., 2016*; *Phillips et al., 2019*).

This study focused on ventral hippocampal projections to the LS (*Risold and Swanson, 1997*). The LS, a subcortical structure primarily composed of GABAergic neurons (*Risold and Swanson, 1997*; *Reid et al., 2024*; *Simon et al., 2024*), receives dense projections from the vHPC and is well situated to orchestrate social novelty-related approach behaviors (*Gergues et al., 2020*; *Besnard and Leroy, 2022*; *Rizzi-Wise and Wang, 2021*; *Wirtshafter and Wilson, 2021*; *Menon et al., 2022*). For instance, LS neurons in rat pups differentially respond to sibling and non-sibling odors and are necessary for the expression of non-sibling preference in older pups (*Clemens et al., 2020*). Also, disrupting vasopressin receptor function in the LS modulates the ability of rodents to discriminate between novel and familiar conspecifics (*Bielsky et al., 2005*; *Bychowski et al., 2013*). Most importantly, the LS is thought to directly project to the VTA (*Luo et al., 2011*), a region previously implicated in social novelty-related behaviors (*Smith et al., 2017*; *Solié et al., 2022*; *Gunaydin et al., 2014*; *Bariselli et al., 2018*; *Bian et al., 2022*; *Shan et al., 2023*). Thus, the lateral septum is well suited to act on social memory-related information from the ventral hippocampus to, in turn, regulate motivated social behaviors.

To determine if the vHPC-LS circuit is causally involved in preferentially investigating a novel conspecific over a familiar conspecific, we chemogenetically silenced vHPC-LS neurons and found that mice spent equal amounts of time investigating a novel and familiar conspecific. Furthermore, we found that optogenetically inhibiting the activity of vHPC-LS neurons in a temporally and spatially specific manner caused the mice to investigate the conspecific paired with the inhibition. We hypothesized that this increased social investigation with vHPC-LS inhibition might arise from disinhibition of the VTA. Using monosynaptic rabies tracing technology, we determined that LS-VTA neurons receive dense ventral hippocampal inputs. Consistent with our hypothesis, we found that chemogenetically silencing LS-VTA neuron activity caused mice to no longer prefer the novel conspecific. Finally, using monosynaptic rabies tracing experiments in Th-Cre[+] mice, we found that the LS makes direct monosynaptic connections onto dopaminergic neurons in the VTA. Taken together, we have identified a vHPC-LS-VTA circuitry that appears to play a key role in transforming social memory-related information in the ventral hippocampus to subsequently drive social novelty-related approach behaviors.

## Results

### Inhibiting vHPC-LS neurons disrupts the preference for social novelty

To determine whether vHPC-LS neurons played a causal role in allowing animals to discriminate between novel and familiar mice, we chemogenetically silenced vHPC-LS neurons while evaluating the effects on behavior in the social discrimination task (SDT). To selectively silence vHPC-LS neurons, we injected C57BL/6J mice in the LS with a retrogradely transporting Cre virus and an AAV5 virus expressing Cre-dependent inhibitory DREADD (hM4Di)(*Armbruster et al., 2007*) tagged with mCherry (or a control virus expressing only mCherry) in the vHPC (*Figure 1A and B*). Three weeks after surgery, we injected mice (i.p.) with either saline or CNO 30 min prior to the SDT (*Figure 1C*). In the SDT, mice were allowed to explore an arena containing age- and sex-matched novel and familiar conspecifics (*Figure 1C*). Typically, mice spend more time in the proximity of a novel conspecific relative to the familiar conspecific (*Moy et al., 2004*). We found that chemogenetic inhibition of vHPC-LS neurons by administration of CNO but not saline in hM4Di expressing mice (male and female) disrupted the

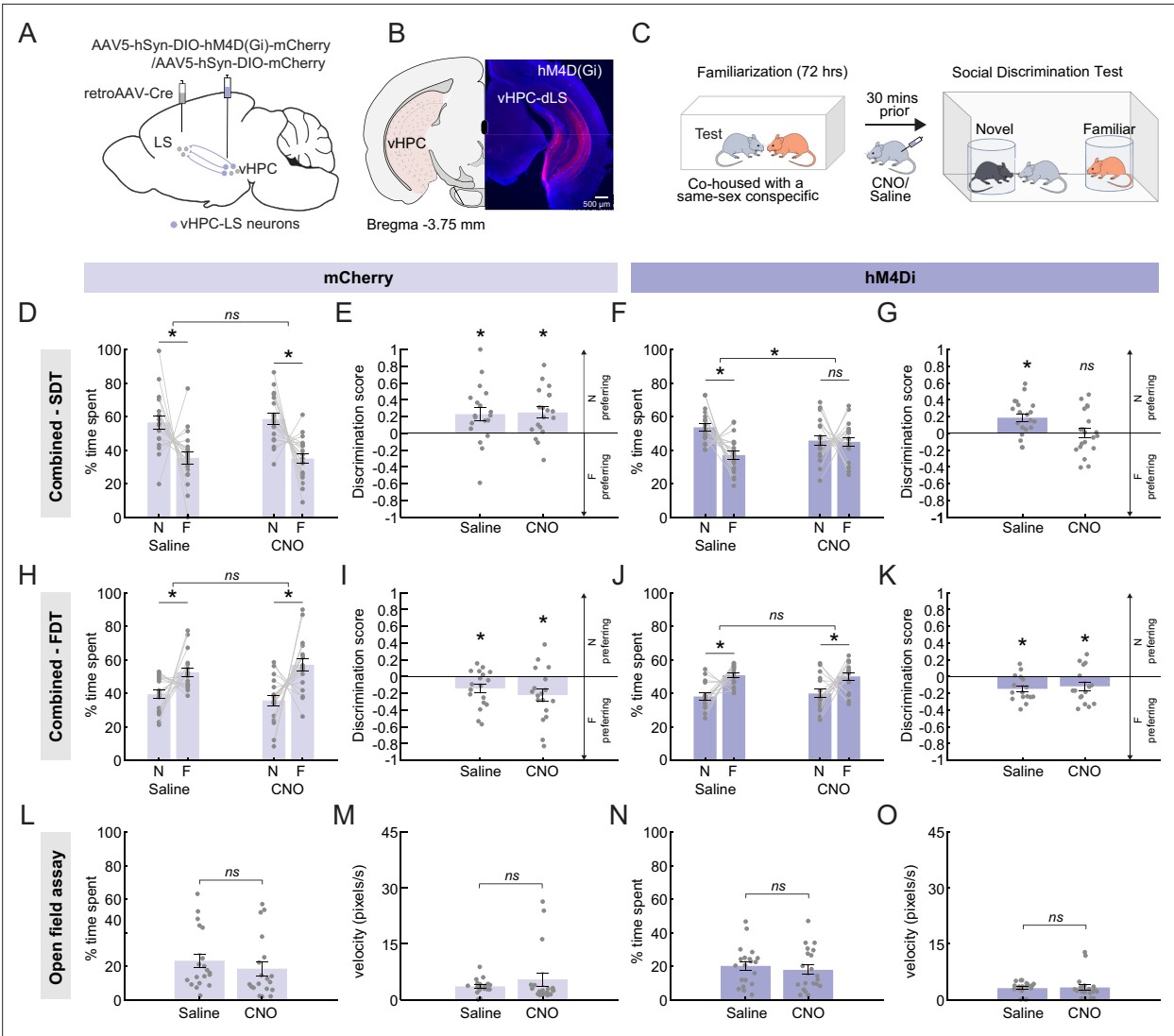

**Figure 1.** Chemogenetic inhibition of ventral hippocampus (vHPC)-lateral septum (LS) disrupts social novelty preference. (**A**) A schematic showing AAV5-hSyn-DIO-hM4D(Gi)-mCherry injection in the vHPC and retroAAV-Cre injection in the LS. (**B**) Example histology showing hM4Di-mCherry expression in vHPC-LS neurons. (**C**) After being pair housed for 72 hr with a sex-matched and age-matched conspecific for familiarization, mice are run through the social discrimination task (SDT). In the task, mice can freely explore an arena containing two encaged conspecifics, one novel and one familiar. (**D**) Control mice expressing the mCherry-only virus in vHPC-LS neurons were injected with either saline (left) or CNO (right) prior to being run on the SDT. Control mice, regardless of treatment group, preferentially spent more time in the proximity of the novel conspecific relative to the familiar conspecific (Two-factor ANOVA with drug condition (saline or CNO) and conspecific identity (novel or familiar) as factors; interaction: p=0.638, main effect of conspecific identity: p=6.4E-06, post hoc Sidak multiple comparison tests; Saline: p=1.1E-04, CNO: p=1.9E-05; mCherry: n=19 mice). (**E**) Discrimination scores show that control mice under both saline and CNO conditions preferentially spend more time investigating the novel conspecific relative to familiar conspecific (One sample t-test, Saline: p=0.0103, CNO: p=0.0013, mCherry: n=19 mice). (**F, G**) Chemogenetic inhibition of vHPC-LS neurons with CNO disrupted the preference of mice for novel conspecific in the SDT. In contrast, hM4Di expressing mice exhibited a strong preference for the novel conspecific over the familiar conspecific when mice were administered saline prior to being run on the SDT (Two-factor ANOVA with drug condition and conspecific identity as factors; interaction: p=8.4E-04, main effect of conspecific identity: p=0.003; post hoc Sidak multiple comparison tests; Saline: p=2E-05, CNO: p=0.971; discrimination score: one sample t-test, Saline: p=0.0009, CNO: p=0.9222; hM4Di: n=20 mice). (**H, I**) Control mice expressing the mCherry-only virus in vHPC-LS neurons were injected with either saline (left) or CNO (right) prior to being run on a food discrimination task. Control mice preferentially spent more time in the proximity of familiar food regardless of the drug treatment group (Two-factor ANOVA with food identity (novel or familiar) and drug condition as factors; interaction: p=0.254, main effect of food identity: p=0.003, post hoc Sidak multiple comparison tests; Saline: p=0.0323, CNO: p=0.0009; discrimination score: one sample t-test, Saline: p=0.0134, CNO: p=0.008; mCherry: n=18 mice). (**J, K**) Importantly, inhibition of hM4Di expressing vHPC-LS neurons in the presence of CNO had no effect on the ability of mice to preferentially investigate the familiar food relative to the novel food. Both CNO- and saline-injected hM4Di animals preferentially spent more time in the proximity of the familiar food (Two-factor ANOVA with food identity and drug condition as factors; interaction: p=0.508, main effect of food preference: p=3.2E-05, post hoc

*Figure 1 continued on next page*

*Figure 1 continued*

Sidak multiple comparison tests; Saline: p=1.7E-04, CNO: p=0.03; discrimination score: one sample t-test, Saline: p=0.0016, CNO: p=0.0419; hM4Di: n=16 mice). (**L, N**) We observed that both saline- and CNO-injected control (**L**) and hM4Di (**N**) mice spent similar amounts of time in the center of the open field arena (paired t-test, mCherry: p=0.3167, hM4Di: p=0.1837; mCherry: n=19 mice, hM4Di = 20 mice). (**M, O**) In both mCherry (**M**) and hM4Di (**O**) mice, saline and CNO injections did not affect the velocity (pixel/second) of the animals (paired t-test, mCherry: p=0.2694, hM4Di: p=0.7886; mCherry: n=19 mice, hM4Di: n=20 mice). All error bars denote standard error of the mean.

The online version of this article includes the following source data and figure supplement(s) for figure 1:

**Source data 1.** Data associated with *Figure 1*.

**Figure supplement 1.** Chemogenetic inhibition of ventral hippocampus (vHPC)-lateral septum (LS) disrupts social novelty preference in both male and female mice.

ability of mice to preferentially spend more time in the proximity of the novel animal (*Figure 1F*; two-factor ANOVA with drug condition and conspecific identity as factors; interaction: p=8.4E-04, main effect of conspecific identity: p=0.003; post hoc Sidak multiple comparison tests: Saline: p=2E-05, CNO: p=0.971; *Figure 1 - Figure Supplement 1*). CNO administration in mCherry-expressing control mice did not affect their natural preference for the novel conspecific (*Figure 1D*; two-factor ANOVA with drug condition and conspecific identity as factors; interaction: p=0.638, main effect of conspecific identity: p=6.40E-06; post hoc Sidak multiple comparison tests: Saline: p=1.10E-04, CNO: p=1.90E-05). In both saline and CNO conditions, the discrimination scores of the mCherry mice show that mice preferentially spend more time investigating the novel conspecific relative to familiar conspecific (*Figure 1E*; one sample t-test, Saline: p=0.0103, CNO: p=0.0013) while inhibition of vHPC-LS neurons disrupts the preference for the novel mouse (*Figure 1G*; one sample t-test, Saline: p=0.0009, CNO: p=0.9222).

To determine if the effects observed with vHPC-LS inhibition are socially specific, we ran the mice through a food discrimination assay in which mice were allowed to explore an arena containing novel and familiar foods. Unlike with conspecifics, mice are neophobic and do not prefer novel foods (*File, 2001*). Therefore, we expected mice would spend less time around the novel food in comparison to the familiar food. All mice exhibited a preference for the familiar food regardless of virus or drug condition (*Figure 1H–K*; two-factor ANOVA with food identity and drug condition as factors; mCherry: interaction: p=0.254, main effect of food preference: p=0.003, post hoc Sidak multiple comparison tests: Saline: p=0.0323, CNO: p=9E-04; discrimination score: one sample t-test, Saline: p=0.0134, CNO: p=0.008; hM4Di: interaction: p=0.508, main effect of food preference: p=3.2E-05, post hoc Sidak multiple comparison tests: Saline: p=1.7E-04, CNO: p=0.003; discrimination score: one sample t-test, Saline: p=0.0016, CNO: p=0.0419). This finding suggests that the effects observed with vHPC-LS inhibition are socially specific.

To determine if the reduction in preference for a novel conspecific in the SDT could arise from anxiety-like behaviors, we ran mice through an open field assay. We found that chemogenetic silencing of vHPC-LS neurons had no effect on the time spent in the center of the arena (*Figure 1L and N*; paired t-test, mCherry: p=0.3167, hM4Di: p=0.1837). Additionally, chemogenetic inhibition of vHPC-LS neurons had no effect on locomotion, as evidenced by the absence of any changes in the velocity of the animals (*Figure 1M and O*; paired t-test, mCherry: p=0.2694, hM4Di: p=0.7886). Thus, ruling out the possibility that altered locomotion could be contributing to the ability of mice to discriminate between novel and familiar conspecifics.

Taken together, vHPC-LS chemogenetic inhibition disrupted the ability of mice to preferentially investigate a novel conspecific over a familiar conspecific while not affecting their ability to discriminate between novel and familiar foods. However, it is unclear whether chemogenetic inhibition increases the preference for a familiar mouse or decreases preference for a novel mouse. To distinguish between these possibilities, we next optogenetically silenced the activity of vHPC-LS neurons in a temporally and spatially restricted manner.

## vHPC-LS neuron inhibition increases investigation of a mouse paired with inhibition

To determine whether inhibition of vHPC-LS neurons differentially influences the investigation of a novel versus a familiar mouse, we used halorhodopsin (NpHR) to silence vHPC-LS neurons in

a temporally and spatially specific manner. We injected wild-type mice with a retrogradely transporting Cre virus in the LS and a Cre-dependent NpHR-EYFP or EGFP-only virus in the vHPC and then implanted fibers in the vHPC to target vHPC-LS cell bodies (*Figure 2A*, *Figure 2 - Figure Supplement 1*). Using a closed-loop optogenetic manipulation (See Methods), we selectively inhibited vHPC-LS neurons when a conspecific entered a previously defined social zone (*Figure 2C*).

Interestingly, we found that inhibition of vHPC-LS neurons in the proximity of a familiar mouse abolished the natural preference mice have for novel conspecifics (*Figure 2F and G*; two-factor ANOVA with light condition and conspecific identity as factors; interaction: p=0.024, main effect of conspecific identity: p=0.006, Sidak post hoc test; OFF: p=0.006, N-ON: p=0.037, F-ON: p=0.949; discrimination score: one sample t-test, OFF: p=0.002, NON: p=0.19, FON: p=0.719). Significantly, light stimulation did not affect the preference for the novel mouse in the control-EGFP group (*Figure 2D and E*; two-factor ANOVA with light condition and conspecific identity as factors; interaction: p=0.898, main effect of conspecific identity: p=2.8E-05, Sidak post hoc test; OFF: p=0.032, N-ON: p=0.012, F-ON: p=0.032; discrimination score: one sample t-test, OFF: p=0.008, NON: p=0.009, FON: p=0.028). These findings raise the possibility that inhibiting vHPC-LS neurons in the proximity of the familiar mouse could drive increased investigation of that mouse, thus decreasing the time spent in the proximity of a novel mouse.

To determine if silencing of vHPC-LS neurons increased investigation of the mouse paired with stimulation, we inhibited vHPC-LS neurons of test mice in the proximity of one of two novel mice in an arena (*Figure 2H–K*). We found that NpHR-injected mice spent more time in the proximity of the novel mouse that was paired with inhibition than the other novel animal (*Figure 2J and K*; two-factor ANOVA with light condition and conspecific identity as factors; interaction: p=0.040, main effect of conspecific identity p=0.005; post hoc Sidak tests; N-OFF: p=0.961, N-ON: p=0.001; discrimination score: one sample t-test, OFF: p=0.796, NON: p=0.029). The control-EGFP group showed an equal preference for both novel animals regardless of the light condition (*Figure 2H, I*; two-factor ANOVA with light condition and conspecific identity as factors; interaction: p=0.006, no main effect of conspecific identity: p=0.608; discrimination score: one sample t-test, OFF: p=0.535, NON: p=0.128).

Importantly, we found that the increased investigation observed with inhibition of vHPC-LS neurons was socially specific. Inhibition of vHPC-LS activity in the proximity of a novel object did not increase the time spent in the proximity of the object (*Figure 2N and O*; two-factor ANOVA with light condition and object identity as factors; interaction: p=0.871, no main effect object identity: p=0.505; discrimination score: one sample t-test, OFF: p=0.720, NON: p=0.905). Light stimulation also did not change preferences in the control-EGFP group when presented with two novel objects (*Figure 2L and M*; two-factor ANOVA with light condition and object identity as factors; interaction: p=0.878, no main effect of object identity: p=0.220; discrimination score: one sample t-test, OFF: p=0.697, N-ON: = 0.422). These findings suggest that the increased investigation of a mouse paired with vHPC-LS inhibition was not simply a result of the inhibition being appetitive.

To determine if changes in locomotion or anxiety-related behaviors could drive increased investigation of the mouse paired with inhibition, we compared both the time spent in the center and velocity of the animals with and without inhibition of the vHPC-LS neurons while mice explored an open field. We found that inhibition of vHPC-LS neurons did not affect either the velocity or the time spent in the center of the arena in the open field assay (*Figure 2R and S*; one-way ANOVA, time spent: p=0.254; velocity: p=0.775). Light stimulation also had no effect on time spent in the center of the open field arena and the velocity of the animals in the control-EGFP group (*Figure 2P and Q*; one-way ANOVA, time spent: p=0.094; velocity: p=0.259).

Together, these results suggest that vHPC-LS inhibition likely disrupts preference for a novel conspecific by increasing investigation of a familiar conspecific. Thus, demonstrating that intact activity in the vHPC-LS pathway is necessary for mice to preferentially engage in social novelty-related behaviors. These findings raise the possibility that inhibition of the vHPC-LS neurons could disinhibit downstream regions that have been implicated in promoting approach of novel conspecifics, like the VTA (*Smith et al., 2017*; *Solié et al., 2022*; *Gunaydin et al., 2014*; *Bariselli et al., 2018*; *Bian et al., 2022*; *Shan et al., 2023*), by decreasing the inhibitory drive that GABAergic LS projection neurons exert on their downstream targets.

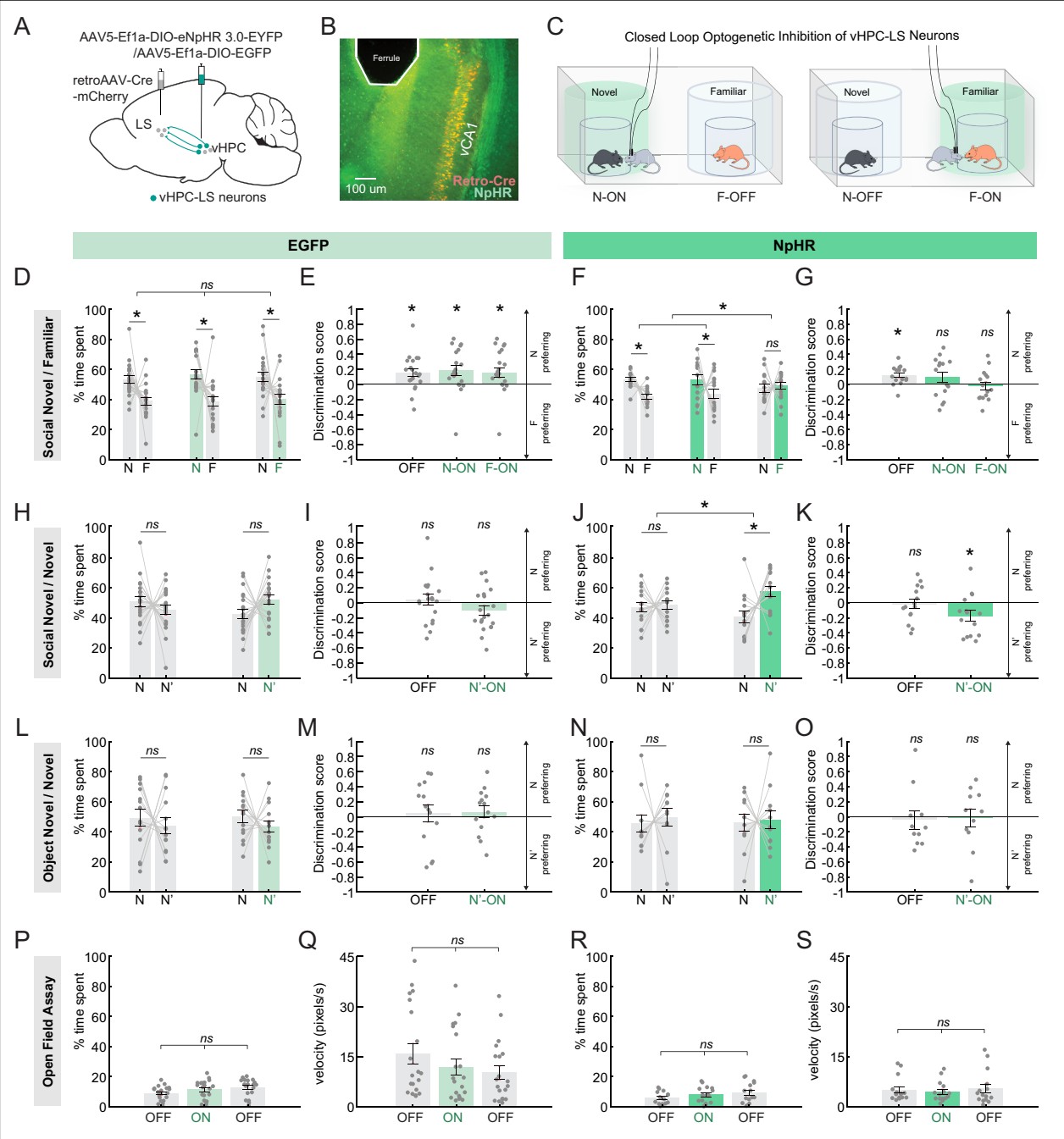

**Figure 2.** Ventral hippocampus (vHPC)-lateral septum (LS) neuron inhibition increases investigation of a mouse paired with inhibition. (**A**) A schematic showing retroAAV-Cre-mCherry injection in the LS and AAV5-Ef1a-DIO eNpHR 3.0-EYFP injection in the vHPC. Optical ferrules were implanted in the vHPC to inhibit halorhodopsin (NpHR) expressing vHPC-LS neurons. (**B**) Example histology showing vHPC-LS neurons labeled with NpHR (expressing both EYFP and mCherry) and ferrule placement. (**C**) As with the chemogenetic condition, mice were pair housed for 72 hr with a sex-matched, and age-matched conspecific for familiarization, then are run through the social discrimination task (SDT). In the task, mice can freely explore an arena containing two encaged conspecifics, one novel and one familiar. vHPC-LS neurons were inhibited (532 nm; 6 mW; constant light stimulation) in the proximity of one of the two conspecifics. (**D**) Control mice expressing EGFP had light stimulation paired with either a novel (N–ON) or a familiar conspecific (F–ON). We also had a stimulation-free condition (OFF). Control mice, regardless of stimulation group, preferentially spent more time in the proximity of the novel conspecific relative to the familiar conspecific (Two-factor ANOVA with light condition (on or off) and conspecific identity as factors; interaction: p=0.898, main effect of conspecific identity: p=2.8E-05; post hoc Sidak multiple comparison tests; OFF: p=0.032; N-ON: p=0.012; F-ON: p=0.032; EGFP: n=20 mice). (**E**) Discrimination scores show that control mice in all light stimulation conditions preferentially spend more time investigating the novel conspecific relative to familiar conspecific (One sample t-test, OFF: p=0.008, N-ON: p=0.009, F-ON: p=0.028; EGFP: n=20 mice). (**F, G**) Mice expressing NpHR had light stimulation paired with a novel (N–ON) or familiar conspecific (F–ON), in addition to a stimulation-free condition (OFF). NpHR mice

*Figure 2 continued on next page*

*Figure 2 continued*

preferentially spent more time in the proximity of the novel conspecific relative to the familiar conspecific, except when stimulation was paired with a familiar conspecific (Two-factor ANOVA with light condition and conspecific identity as factors; interaction: p=0.024, main effect of conspecific identity p=0.006, post hoc Sidak multiple comparison tests; OFF: p=0.006, N-ON: p=0.037, F-ON: p=0.949, discrimination score: one sample t-test, OFF: p=0.002, N-ON: p=0.19, F-ON: p=0.719; NpHR: n=15 mice). (**H, I**) EGFP mice were then run through the SDT but with two novel conspecifics to look at the impact of stimulation on novelty preference. Mice were run in either light off (OFF) condition or stimulated when in the proximity of one of two novel animals (N'-ON). Control mice, regardless of stimulation group, spent an equivalent amount of time in the proximity of each novel conspecific (Two-factor ANOVA with light condition and conspecific identity as factors; interaction: p=0.006, no main effect of conspecific identity: p=0.608; post hoc Sidak multiple comparison tests; OFF: p=0.376, N-ON: p=0.075; discrimination scores: one sample t-test, OFF: p=0.535, N-ON: p=0.128; EGFP: n=20 mice) (**J, K**) NpHR mice were then run through the novel SDT in the light off or N'-ON condition. NpHR mice exhibited a preference for the novel conspecific paired with vHPC-LS inhibition (Two-factor ANOVA with light condition and conspecific identity as factors, interaction: p=0.040, main effect of conspecific identity: p=0.005, post hoc Sidak multiple comparison tests; N-OFF: p=0.961, N-ON: p=0.001; discrimination scores: one sample t-test, OFF: p=0.796, N'-ON: p=0.029; NpHR: n=20 mice) (**L, M**) Control mice, regardless of stimulation group, showed an equal preference for both novel objects (Two-factor ANOVA with light condition and object identity (N or N') as factors; interaction: p=0.878, no main effect of object identity: p=0.220, post hoc Sidak multiple comparison tests; N-OFF: p=0.704, N-ON: p=0.556; discrimination score: one sample t-test, OFF: p=0.697, N-ON: p=0.422; EGFP: n=14 mice) (**N, O**) vHPC-LS inhibition had no effect on object preference even when one of the two novel objects were paired with stimulation. (Two-factor ANOVA with light condition and object identity as factors; interaction: p=0.871, no main effect of object identity: p=0.505, post hoc Sidak multiple comparison tests; N-OFF: p=0.853, N-ON: p=0.968; discrimination score: one sample t-test, OFF: p=0.720, N-ON: p=0.905; NpHR: n=11 mice) (**P, R**) We observed that, regardless of light condition, control (**P**) and NpHR (**R**) mice spent similar amounts of time in the center of the open field arena (one-way ANOVA, EGFP: p=0.094, 20 mice; NpHR: p=0.254, 15 mice). (**Q, S**) We observed that light conditions had no effect on the speed (pix/sec) of control (**Q**) and NpHR (**S**) mice (One-way ANOVA, EGFP: p=0.259, 20 mice; NpHR: p=0.775, 15 mice). Green bars denote light ON condition and gray bars denote light OFF condition. All error bars denote standard error of the mean.

The online version of this article includes the following source data and figure supplement(s) for figure 2:

**Source data 1.** Data associated with *Figure 2*.

**Figure supplement 1.** Histological reconstruction of optical fiber tip placements.

## LS-VTA neurons receive dense monosynaptic input from the vHPC

We hypothesized that effects observed with vHPC-LS inhibition on social novelty-related approach behaviors could be mediated via LS projections to the VTA. To test this hypothesis, we first had to establish the existence of the vHPC-LS-VTA circuitry. To confirm that LS neurons do project to the VTA, we injected a retrogradely transported Cre virus into the VTA and a Cre-dependent GFP virus into the LS (*Figure 3A and B*). From dense GFP labeling in the LS, we were able to confirm that LS neurons projected to the VTA (*Figure 3C*).

Next, we determined if LS-VTA neurons received monosynaptic inputs from the vHPC using a modified monosynaptic rabies tracing technique (*Wickersham et al., 2007*). Three weeks after injecting a retrogradely transporting Cre virus in the VTA and a Cre-dependent rabies helper virus in the LS, we injected the modified delta G-deleted rabies virus into the LS (*Figure 3D–F*). Using the semi-automated WholeBrain software (*Fürth et al., 2018*), we mapped brain-wide monosynaptic inputs onto LS-VTA neurons (*Figure 3G*). We quantified the proportion of total neurons from each region that directly synapse onto LS-VTA neurons. The hippocampal formation was the largest source of inputs to LS-VTA neurons (*Figure 3H*; one-way ANOVA: p=0.0107, Post hoc multiple comparison Dunnett's test; HPF vs IC: p=0.1579, HPF vs HY: p=0.0206, HPF vs TH: p=0.0056). Within the hippocampus, the CA1 and CA3 neurons projected most densely onto LS-VTA neurons (*Figure 3I*; one-way ANOVA: p=0.0096, Post hoc multiple comparison Dunnett's test; CA1 vs DG: p=0.0411, CA1 vs SUBd: p=0.0511, CA3 vs DG: p=0.0134, CA3 vs SUBd: p=0.0166, CA3 vs SUBv: p=0.0325.). When looking at the organization of inputs along the dorsoventral axis of the hippocampus, we observed that significantly more vCA1 than dCA1 neurons project to LS-VTA neurons (*Figure 3J*; paired t-test: p=0.0496). In contrast, an equivalent proportion of both dCA3 and vCA3 neurons project onto LS-VTA neurons (*Figure 3K*; paired t-test: p=0.7736).

From rabies tracing experiments, we have established that the vHPC neurons, specifically neurons from the vCA1 and vCA3 subregions, make dense and direct monosynaptic connections onto LS neurons that in turn project to the VTA. Thus, establishing the existence of a hippocampal-septal-ventral tegmental area circuit that could play a critical role in shaping social novelty discrimination.

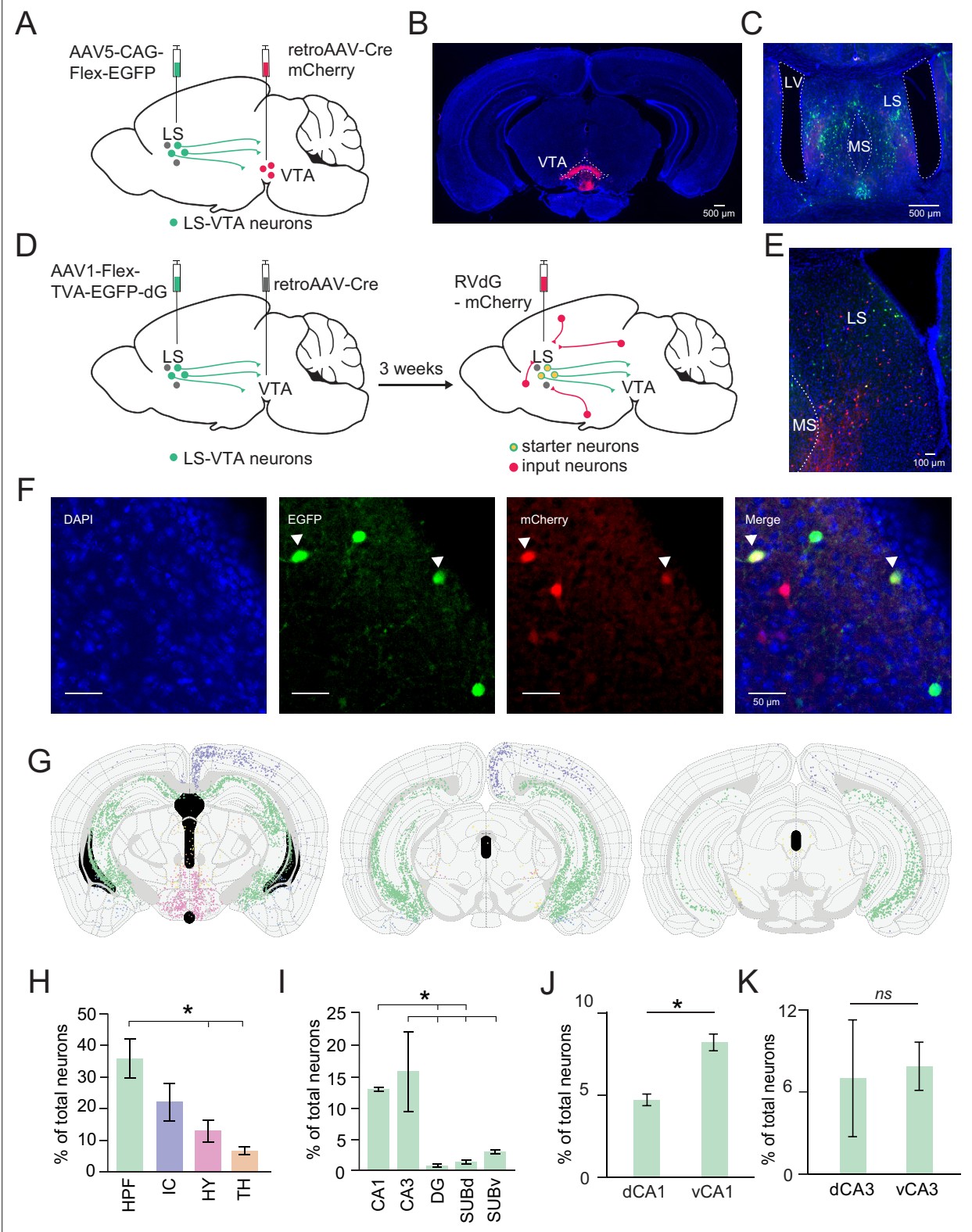

**Figure 3.** Lateral septum (LS)-ventral tegmental area (VTA) neurons receive dense monosynaptic input from the hippocampus. (**A**) A schematic of the viral strategy to determine if LS neurons project to the VTA. The schematic shows AAV5-CAG-flex-EGFP injection into the LS and retroAAV-Cre-mCherry injection into the VTA. (**B**) Example histology showing Retro-Cre-mCherry targeting in the VTA (mCherry). (**C**) Histology showing LS neurons (GFP) that project to the VTA. (**D**) A schematic of the viral tracing strategy for investigating monosynaptic inputs into LS-VTA neurons using a modified

*Figure 3 continued on next page*

*Figure 3 continued*

rabies tracing method. (**E, F**) Example histology showing starter cells expressing both GFP and mCherry in the LS. (**G**) Coronal section showing all detected mCherry-labeled input neurons to LS-VTA neurons along the anterior-posterior axis of the hippocampus (−2.5, −3.0, and −3.5 mm from bregma, left to right). Green dots represent inputs to LS-VTA neurons from the hippocampal formation. Purple: Isocortex; Pink: Hypothalamus; Orange: Thalamus; Yellow: Midbrain. (**H**) Proportion of total neurons from hippocampal formation (HPF), Isocortex (IC), Hypothalamus (HY), and Thalamus (TH). A significantly larger proportion of HPF neurons send input to LS-VTA neurons compared to the HY and TH (One-way ANOVA: p=0.0107, Post hoc multiple comparison test; HPF vs IC: p=0.1579, HPF vs HY: p=0.0206, HPF vs TH: p=0.0056). (**I**) Breakdown of proportion of total neurons from subsections of the hippocampal formation. Amongst the HPF, CA1 and CA3 regions send more inputs to LS-VTA neurons compared to other subregions. (One-way ANOVA: p=0.0096, Post hoc multiple comparison test; CA1 vs DG: p=0.0411, CA1 vs SUBd: p=0.0511, CA3 vs DG: p=0.0134, CA3 vs SUBd: p=0.0166, CA3 vs SUBv: p=0.0325). (**J, K**) Distribution of CA1 (**J**) and CA3 (**K**) inputs onto LS-VTA neurons along the dorsoventral axis of the hippocampus. More vCA1 neurons project to LS-VTA neurons compared to dCA1 (**J**, paired t-test: p=0.0496). A comparable proportion of dCA3 and vCA3 neurons project to LS-VTA (**K**, paired t-test: p=0.7736). n=3 mice. All error bars denote standard error of the mean.

The online version of this article includes the following source data for figure 3:

**Source data 1.** Data associated with *Figure 3*.

## LS-VTA neurons play a role in social discrimination and food discrimination

After establishing that LS-VTA neurons do receive monosynaptic inputs from the vHPC, we next asked if LS projections to the VTA played a causal role in allowing mice to preferentially investigate a novel conspecific. To test this possibility, we chemogenetically silenced LS-VTA neurons while mice explored the SDT arena containing a novel and familiar conspecific. We injected a retrogradely transporting Cre virus in the VTA of C57BL/6 J mice and a Cre-dependent inhibitory (hM4Di) DREADD virus or an mCherry virus in the LS (*Figure 4A and B*). Three weeks after surgery, mice received i.p. injections of either saline or CNO to 30 min prior to being run on the SDT (*Figure 4C*).

Interestingly, inhibition of hM4Di expressing LS-VTA neurons with CNO strongly disrupted the ability of mice to preferentially investigate the novel conspecific in the SDT (*Figure 4F and G*; two-factor ANOVA with drug condition and conspecific identity as factors; interaction: p=5.1E-04, post hoc Sidak tests: Saline: p=0.001, CNO: p=0.091; discrimination score: one sample t-test, Saline: p=0.012, CNO: p=0.208). Importantly, control (mCherry) mice exhibited a strong preference for the novel conspecific over the familiar conspecific in both saline and CNO conditions (*Figure 4D and E*; two-factor ANOVA with drug condition and conspecific identity as factors; interaction: p=0.307, main effect of conspecific identity p=6.8E-04, post hoc Sidak tests; Saline: p=2.2E-04, CNO: p=0.0031; discrimination score: one sample t-test, Saline: p=0.045, CNO:p=0.032).

Surprisingly, we found that, unlike with vHPC-LS inhibition, inhibiting LS-VTA neurons appears to cause a trend towards disrupting the mice's normal preference for familiar foods. Inhibiting the activity of LS-VTA neurons caused mice to trend towards spending equal amounts of time in the proximity of novel and familiar foods (*Figure 4J and K*; two-factor ANOVA with drug condition and food identity as factors; interaction: p=0.076; discrimination score: one sample t-test, Saline:p=0.065, CNO: p=0.863). Control animals preferred the familiar food in both saline and CNO conditions (*Figure 4H, I*; two-factor ANOVA with drug condition and food identity as factors; interaction: p=0.941; main effect of food preference: p=2.7 E-05; discrimination score: one sample t-test, Saline: p=0.007, CNO: p=0.055).

Additionally, we were able to confirm that these effects observed with LS-VTA inhibition were not driven by non-specific effects on anxiety-related behaviors. We observed that CNO administration had no effect on the time spent in the center of the open field arena in both control and hM4Di animals (*Figure 4L and N*; paired t-test, mCherry: p=0.2196, hM4Di: p=0.2182). We also observed that CNO administration did not affect speed of the animals in both control and hM4Di animals (*Figure 4M and O*; paired t-test, mCherry: p=0.0733, NpHR: p=0.7193).

These findings provide compelling evidence that the increased investigation observed with vHPC-LS inhibition is likely mediated via its projection onto the VTA. Dopamine neurons in the VTA show increased activity while mice investigate a novel conspecific and are known to be involved in causally driving increased investigation of a novel conspecific (*Smith et al., 2017*; *Solié et al., 2022*; *Gunaydin et al., 2014*; *Bariselli et al., 2018*; *Bian et al., 2022*; *Shan et al., 2023*). Together, these findings raise the intriguing possibility that silencing vHPC-LS neurons could disinhibit dopamine neurons in the VTA, which, in turn, could drive increased approach and investigation of a conspecific.

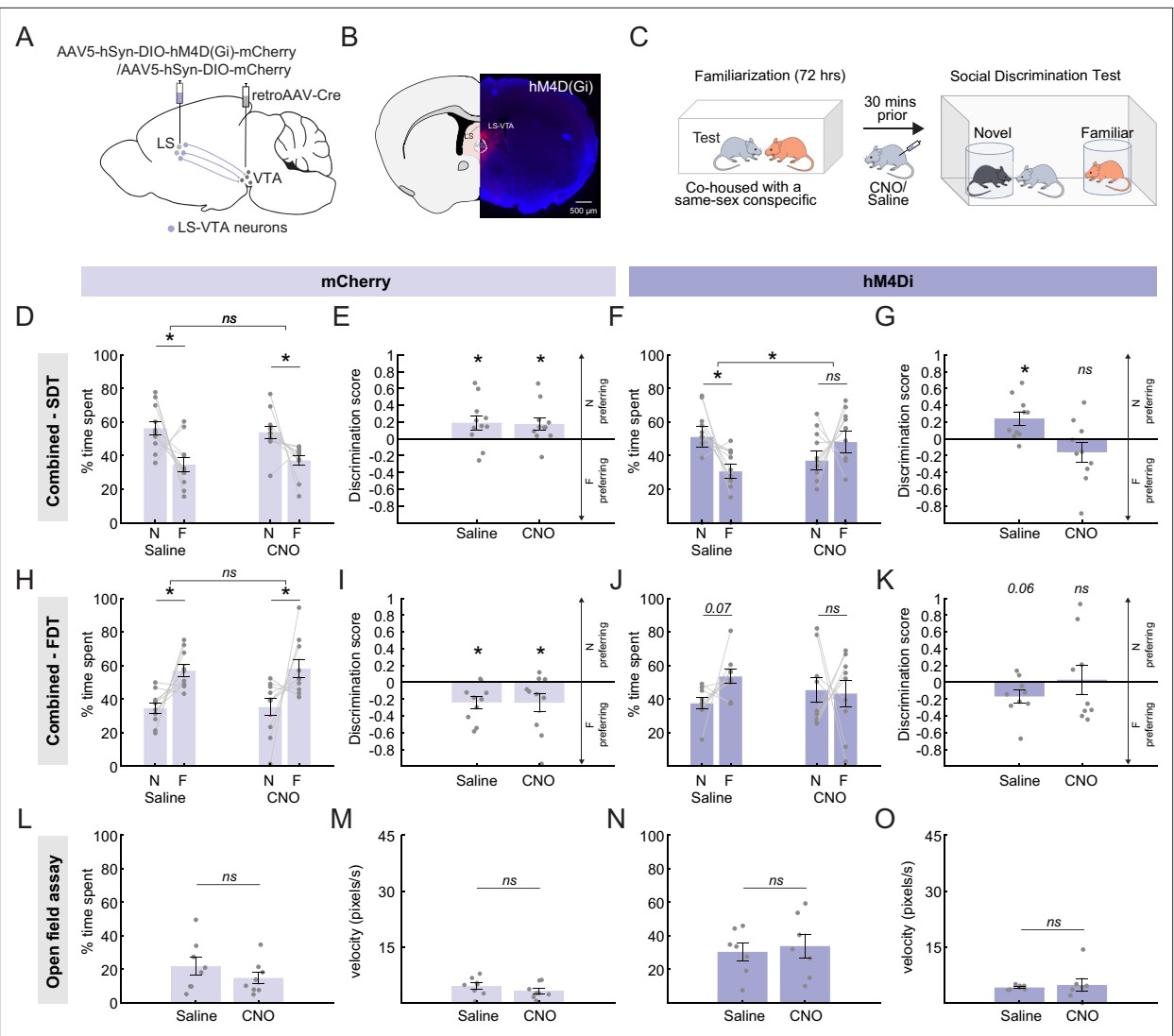

**Figure 4.** Lateral septum (LS)-ventral tegmental area (VTA) neurons play a role in social discrimination and food discrimination. (**A**) A schematic of AAV5-hSyn-DIO-hM4D(Gi)-mCherry injection in the LS and retro AAV-Cre injection in the VTA (**B**) Example histology showing hM4Di-mCherry expression in LS-VTA neurons. (**C**) Mice are pair housed for 72 hr with a sex- and age-matched conspecific for familiarization, and then mice are run through the social discrimination task (SDT). In the task, mice are allowed to freely explore an arena containing two encaged conspecifics, one novel and one familiar. (**D, E**) Control mice expressing the mCherry-only virus in LS-VTA neurons were injected with either saline (left) or CNO (right) prior to being run on the SDT. Control mice, regardless of treatment group, preferentially spent more time in the proximity of the novel conspecific relative to the familiar conspecific (Two-factor ANOVA with drug condition and conspecific identity as factors; interaction: p=0.307, main effect of conspecific identity p=6.8E-04, post hoc Sidak multiple comparison tests; Saline: p=2.2E-04, CNO: p=0.003; discrimination score: one sample t-test, Saline: p=0.045, CNO: p=0.032, n=11 mice). (**F, G**) Chemogenetic inhibition of LS-VTA neurons with CNO disrupted the preference of mice for novel conspecific in the SDT. In contrast, hM4Di expressing mice exhibited a strong preference for the novel conspecific over the familiar conspecific when mice when administered saline prior to being run on the SDT (Two-factor ANOVA with drug condition and conspecific identity as factors; interaction: p=5.1E-04, post hoc Sidak multiple comparison tests; Saline: p=0.001, CNO: p=0.091; discrimination score: one sample t-test, Saline: p=0.012, CNO: p=0.208, n=10 mice). (**H, I**) Control mice expressing the mCherry-only virus in LS-VTA neurons were injected with either saline (left) or CNO (right) prior to being run on a food discrimination task. Control mice preferentially spent more time in the proximity of the familiar food regardless of the drug treatment group (Two-factor ANOVA with drug condition and food identity as factors; interaction: p=0.941, main effect of food preference: p=2.7E-05, post hoc Sidak multiple comparison tests; Saline: p=0.002, CNO: p=0.001; discrimination score: one sample t-test, Saline: p=0.007, CNO: p=0.055, n=10 mice) (**J, K**) Interestingly, there is a trend towards disrupted food preference in the hM4Di animals when administered CNO. This was not observed when the animals received saline injections as saline-injected animals preferred the familiar food (Two-factor ANOVA with drug condition and food identity as factors; interaction: p=0.076, post hoc Sidak multiple comparison tests; Saline: p=0.07, CNO: p=0.81; discrimination score: one sample t-test, Saline: p=0.065 CNO: p=0.863, n=9 mice). (**L, N**) CNO administration did not affect the time spent in the center of the open field arena in both mCherry and hM4Di mice (paired t-test, mCherry: p=0.2196; hM4Di: p=0.2182; mCherry n=8 mice, hM4Di n=7 mice). (**M, O**) Velocity (pixel/seconds) of control and hM4Di mice were unaltered by CNO

*Figure 4 continued on next page*

*Figure 4 continued*

administration (paired t-test, mCherry: p=0.0733, hM4Di: p=0.7193; mCherry n=8 mice, hM4Di n=7 mice). All error bars denote standard error of the mean.

The online version of this article includes the following source data for figure 4:

**Source data 1.** Data associated with *Figure 4*.

Although we have established LS projects to the VTA, it is unclear if LS projects to the dopaminergic neurons in the VTA.

## LS neurons synapse directly onto dopamine neurons in the VTA

To determine if LS neurons synapse directly onto dopamine neurons in the VTA, we mapped monosynaptic inputs onto dopaminergic neurons in the VTA (VTA$_{DA}$) by applying the modified rabies tracing method in Th-Cre$^+$ mice. First, we injected a Cre-dependent helper AAV virus into the VTA of Th-Cre$^+$ mice. After a 3 week period, we injected a delta G-deleted rabies virus expressing mCherry into the VTA (*Figure 5A–C*). Using the semi-automated WholeBrain software (*Fürth et al., 2018*), we mapped and registered mCherry-labeled cells along the entire anterior-posterior axis of the LS (*Figure 5D*). These mCherry-labeled cells signify LS neurons that make monosynaptic connections onto VTA$_{DA}$ neurons. We next quantified the anatomical distribution of VTA$_{DA}$ projecting LS neurons in the brain (*Figure 5E–G*). Amongst the various subdivisions within the LS, we found that the rostral subdivision of the LS (LSr) projected most strongly to the VTA (*Figure 5H*; one-way ANOVA: p=5E-06, post hoc Tukey test; LSr vs LSv: p=1.2E-05, LSr vs LSv: p=6E-06, LSc vs LSv: p=0.26). Thus, we have identified a novel pathway that connects LS neurons directly to dopaminergic VTA neurons that could serve as the neural substrate to drive social novelty-related behaviors.

## Discussion

Here, we show that chemogenetically and optogenetically silencing vHPC-LS neurons disrupts the ability of mice to preferentially investigate a novel conspecific. Furthermore, closed-loop, spatially constrained optogenetic inhibition of vHPC-LS neurons increased investigation of the mouse paired with inhibition. These findings led us to hypothesize that inhibiting vHPC-LS neurons could increase preference for a conspecific by disinhibiting the VTA, a region heavily implicated in social novelty and social approach behaviors. Consistent with this hypothesis, through monosynaptic rabies tracing experiments, we found that vHPC-LS neurons project heavily to the VTA. We next discovered that chemogenetic silencing of LS-VTA neurons also disrupted the ability of mice to preferentially engage with a novel conspecific. Finally, using monosynaptic rabies tracing technology in Th-Cre$^+$ mice, we found that dopaminergic neurons in the VTA receive direct monosynaptic inputs from the LS. Given that LS neurons synapse directly onto VTA dopamine neurons, inhibiting vHPC-LS neurons could disinhibit VTA dopamine neurons, thus driving increased social approach and investigation. We have identified a circuit necessary for mice to differentially investigate novel and familiar conspecifics.

Rodents, including mice, behave differently around novel and familiar conspecifics (*de la Zerda et al., 2022*; *Winslow, 2003*). When given a choice, mice preferentially spend more time near a novel mouse than a familiar mouse. In rodents, optogenetic and imaging experiments have identified the vHPC to be causally involved in distinguishing between familiar and novel conspecifics. However, how information about the conspecific identity thought to be primarily encoded in the vHPC (*Okuyama et al., 2016*) is transformed to drive increased approach and investigation of a novel individual remains poorly understood. In particular, it remains unclear which structures downstream of the hippocampus play a role in regulating differential investigation of a novel over a familiar conspecific. The vHPC sends dense projections onto the nucleus accumbens (NAc), medial prefrontal cortex (mPFC) and the LS (*Gergues et al., 2020*). Recent studies have suggested a key role for hippocampal projections to the mPFC and the NAc in discriminating between novel and familiar conspecifics (*Okuyama et al., 2016*; *Phillips et al., 2019*). However, some questions remain. While optogenetic inhibition of vHPC-NAc neurons disrupted the ability of mice to discriminate novel and familiar conspecifics (*Okuyama et al., 2016*), chemogenetic manipulation of vHPC-NAc neurons did not affect preference for a novel conspecific (*Phillips et al., 2019*). Moreover, while chemogenetic manipulation of vHPC-mPFC activity disrupted the formation of social memory, its effects on modulating social preference

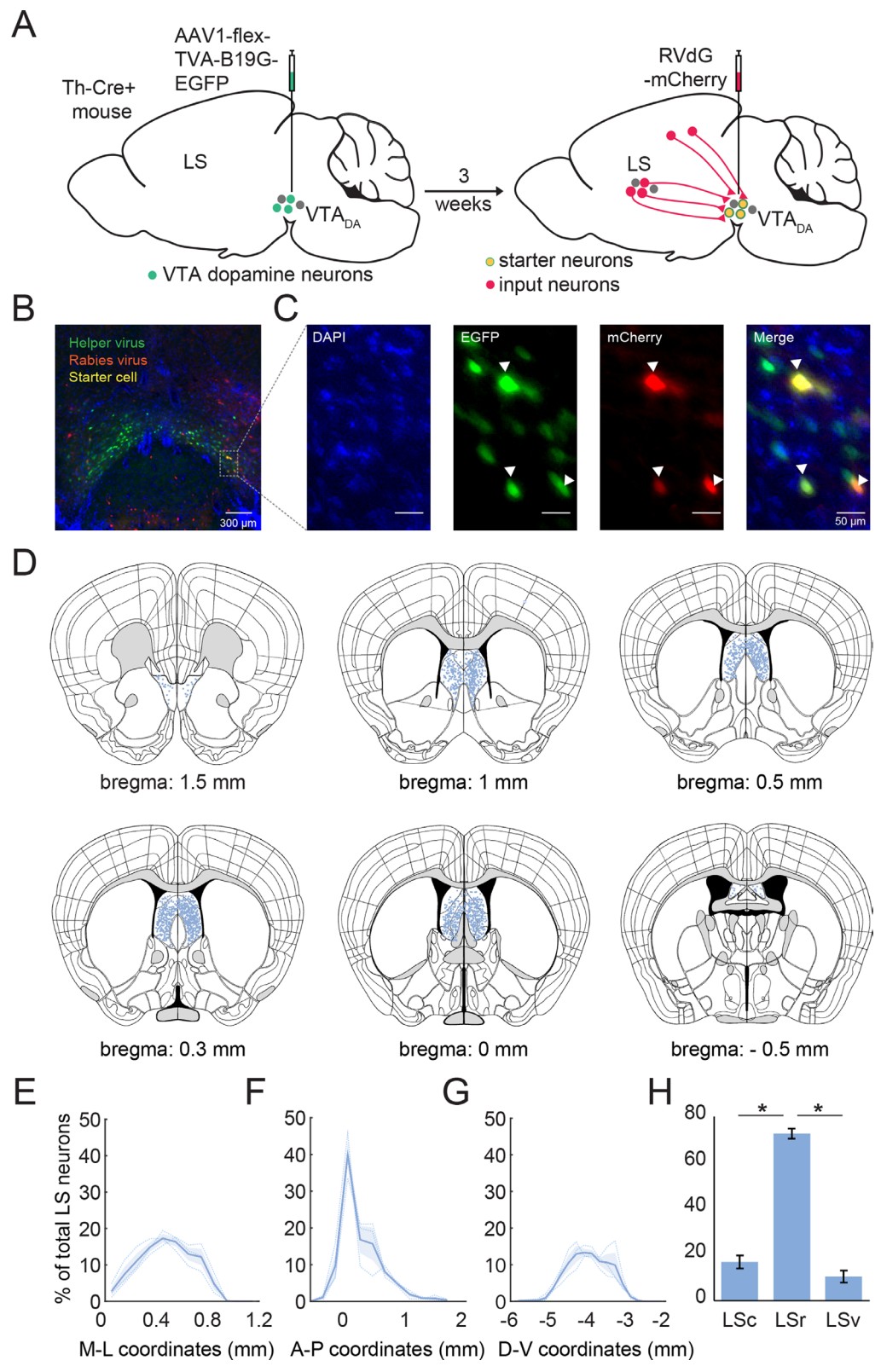

**Figure 5.** Lateral septum (LS) neurons make monosynaptic inputs onto dopaminergic neurons in the ventral tegmental area (VTA). (**A**) Schematic of the viral intersectional strategy for tracing monosynaptic inputs onto dopaminergic neurons in the VTA. A Cre-dependent helper virus (AAV1-synP-FLEX.splitTVA.EGFP.B19G) was first injected into the VTA of Th-Cre+ mice. After allowing 3 weeks for viral expression, a mCherry-labeled, delta

*Figure 5 continued on next page*

*Figure 5 continued*

G-deleted rabies virus was also injected into the VTA (RVdG-mCherry). (**B**) Representative image showing the injection site in the VTA. In green are VTA$_{DA}$ neurons labeled by the helper virus. In red are neurons labeled by the rabies virus. Starter cells that are labeled with both the helper and rabies virus are yellow in this image. (**C**) High resolution (20x) images of the inset showing expression of the TVA delta G helper virus in green and the RVdG-mCherry in red (middle panels). Filled white arrowheads point to starter cells; they are double-labeled and appear yellow (right panel). DAPI stain is in blue (left panel). (**D**) Coronal section showing all detected mCherry-labeled LS input neurons to VTA$_{DA}$ neurons along the anterior-posterior axis of the lateral septum. Each blue dot represents an individual LS neuron that projects to VTA$_{DA}$ neurons. (**E–G**) A normalized density plot along the medio-lateral (**D**; M–L), anterior-posterior (**E**; A–P) and dorsal-ventral axis (**F**; D–V) showing LS neurons that project to dopamine neurons in the VTA. Bin width: 0.1 mm (M–L), 0.2 mm (A–P), 0.2 mm (D–V). (**H**) LS neurons that project onto dopamine neurons in the VTA separated into caudal (LSc), rostral (LSr) and ventral subdivisions (LSv) of the LS. The rostral subdivision of the LS projected most strongly to the VTA (One-way ANOVA: p=5E-06, post hoc Tukey test; LSr vs LSv: p=1.2E-05, LSr vs LSv: p=6E-06, LSc vs LSv: p=0.26, n=3 mice).

The online version of this article includes the following source data for figure 5:

**Source code 1.** Code to generate AP, ML and DV distributions presented in *Figure 5E-G*.

**Source data 1.** Data associated with *Figure 5H*.

**Source data 2.** Data associated with *Figure 5E*.

**Source data 3.** Data associated with *Figure 5F*.

**Source data 4.** Data associated with *Figure 5G*.

for the novel mouse during acute recall in wild-type mice remains less obvious (*Phillips et al., 2019*). Thus, raising the possibility that an alternative pathway from the vHPC could help mice engage in differential social behavior when presented with a novel and familiar conspecific. In our study, using optogenetic and chemogenetic manipulations, we have found that the vHPC-LS pathway plays a key role in modulating preference for a novel conspecific.

vCA1 neurons respond robustly in the proximity of conspecifics and subsets of vCA1 neurons are known to preferentially respond in the proximity of familiar conspecifics (*Okuyama et al., 2016*; *Tao et al., 2022*; *Rao et al., 2019*). Additionally, a larger fraction of vCA1 neurons respond in the proximity of a familiar conspecific compared to a novel conspecific in a social discrimination assay (*Okuyama et al., 2016*). Thus, raising the possibility that lowering the levels of vCA1 neuron activity could create a perception of social novelty and could result in increased investigation of a mouse. Consistent with this idea, we found that optogenetically silencing vHPC-LS neurons increased investigation of the mouse paired with investigation. Findings from our rabies tracing findings show that both vCA1 and vCA3 neurons project to the LS (*Figure 3J and K*). The viral injections in our study were set up to target the vHPC broadly, labeling both vCA1 and vCA3 neurons projecting to the LS. Therefore, the behavioral effects observed with the chemogenetic and optogenetic experiments cannot be localized to only the vCA1 or the vCA3 neurons specifically. A recent study showed that vCA1 and vCA3 neurons differentially modulate approach avoidance conflict task (*Schumacher et al., 2018*). While silencing vCA1 neurons induced increased avoidance of a conflict cue, silencing vCA3 neurons resulted in increased approach of the conflict cue. Additionally, recent data suggests that vCA1 and vCA3 neurons form parallel septal pathways that play differential roles in modulating approach avoidance behaviors in response to conflict cues (*Yeates et al., 2022*). Given that vHPC-LS inhibition did not have any effect on object (*Figure 2N and O*) or food preference (*Figure 1J and K*) in our study, it is unclear if the same vHPC-LS circuits involved in conflict approach/avoidance are also involved in mediating the preference and approach elicited by a novel conspecific. Thus, future studies specifically targeting and manipulating only vCA1 or vCA3 inputs to the LS are necessary to determine if these parallel pathways play differential roles in mediating preferential investigation of a novel conspecific over a familiar conspecific.

The LS, with connections to downstream regions that are heavily implicated in mediating social behaviors, like the NAc (*Dai et al., 2022*; *Williams et al., 2020*; *Le Merrer et al., 2024*; *Park et al., 2021*; *Dölen et al., 2013*; *Walsh et al., 2018*), VTA (*Smith et al., 2017*; *Solié et al., 2022*; *Gunaydin et al., 2014*; *Bariselli et al., 2018*; *Bian et al., 2022*; *Shan et al., 2023*; *Hung et al., 2017*; *O'Connell and Hofmann, 2011*) and ventromedial hypothalamic nucleus (vmH) (*Risold and Swanson, 1997*; *Mei et al., 2023*; *Lin et al., 2011*; *Falkner et al., 2020*; *Lee et al., 2014*; *Hashikawa et al., 2017*), is

well positioned to translate the social recognition-related information it receives from the vHPC into motivated behaviors. In this study, we focused on the LS projections to the VTA, a region that has been shown to play a key role in modulating social novelty. Dopamine neurons in the VTA have been shown to be differentially modulated by novel and familiar conspecifics (*Solié et al., 2022*; *Gunaydin et al., 2014*; *Molas et al., 2017*; *Tapper and Molas, 2020*; *Akiti et al., 2022*). Chemogenetic inhibition of dopamine neurons in the VTA attenuates investigation of a novel conspecific (*Bariselli et al., 2018*). We, therefore, hypothesized that the LS could regulate social novelty preference by modulating the activity of dopamine neurons in the VTA. Using transsynaptic rabies tracing, we identified direct monosynaptic inputs from the LS onto dopamine neurons in the VTA (*Figure 5*). Importantly, we found that inhibiting LS-VTA neurons disrupted the ability of mice to preferentially investigate a novel conspecific.

Unlike the socially specific effects observed with vHPC-LS inhibition, LS-VTA inhibition also disrupted the natural preference that mice exhibit towards investigating familiar foods (*Figure 4J and K*). These findings suggest that LS-VTA neurons are more broadly involved in mediating preference for a variety of ethologically relevant stimuli. Consistent with this idea, VTA dopamine neurons are known to play a key role in mediating the motivation to seek both food rewards and social investigation (*Solié et al., 2022*; *Gunaydin et al., 2014*; *Bariselli et al., 2018*; *Schultz et al., 1997*; *Willmore et al., 2023*; *Cohen et al., 2012*; *Mazzone et al., 2020*). Additionally, a large proportion of individual VTA$_{DA}$ neurons respond to both food and social stimuli (*Willmore et al., 2023*). VTA$_{DA}$ neurons are active during both the consumption of familiar palatable food and during increased investigation of a novel conspecific (*Solié et al., 2022*; *Gunaydin et al., 2014*; *Bariselli et al., 2018*; *Schultz et al., 1997*; *Willmore et al., 2023*; *Cohen et al., 2012*; *Mazzone et al., 2020*). Thus, inhibiting LS neurons that project to the VTA (including VTA$_{DA}$ neurons) could disrupt both social and food-seeking behaviors.

Although our data support the view that inhibition of LS-VTA neurons could be exerting an effect on social novelty preference behaviors by directly modulating the activity of dopamine neurons in the VTA, it remains unknown if LS-VTA neurons also act on GABAergic neurons in the VTA. Local infusions of GABA into the caudodorsal LS was shown to both increase and decrease the responses of VTA neurons in response to stimulation of vCA3 neurons, raising the possibility that LS neurons could project to GABAergic neurons in the VTA (*Luo et al., 2011*). However, it remains to be determined if GABAergic neurons in the VTA receive direct monosynaptic input from the LS. Additionally, future recording and imaging experiments are needed to determine what information LS-VTA neurons encode. For instance, sibling and non-sibling odor representations are anatomically organized along the dorsoventral axis of the LS neurons in rat pups and lesions to the LS disrupt the ability of rat pups to distinguish between kin and non-kin odors (*Clemens et al., 2020*). These findings raise two interesting questions. First, are social familiarity and novelty information differentially represented in the dorsal and ventral populations of LS neurons? Second, are the dorsally located LS-VTA neurons (*Figure 3C*) necessary for kin recognition? Addressing these questions might provide insight into whether different kinds of recognition information (kin/non-kin, novel/familiar) and context information are integrated and transformed into approach and avoidance behaviors at the level of the LS.

In line with the above ideas, neurons from other hippocampal compartments, besides the vCA1 and vCA3, including the dCA3 (*Figure 3K*) and dCA2 (*Leroy et al., 2018*; *Hashimoto et al., 2022*) project to the lateral septum. These hippocampal projections have also been implicated in mediating social novelty behaviors. For example, activating dCA3-LS neurons was sufficient to restore social novelty preference deficits induced by chronic social defeat stress (*Liu et al., 2022*). Interestingly, another study found that silencing the vCA3 neurons but not dCA3 neurons disrupted the ability of mice to differentially investigate familiar versus novel animals in control mice (not defeated/stressed) (*Chiang et al., 2018*). This raises the intriguing possibility that, first, social recognition information is represented in a distributed fashion across the various hippocampal compartments and that parallel hippocampal-septal pathways are differentially recruited in a context and internal state-dependent fashion to transform social novelty/familiarity information into decisions to approach or avoid a conspecific.

Mounting evidence suggests a key role for the lateral septum in modulating a broad range of motivated social behaviors. In addition to the role we are suggesting for the vCA1-LS pathway in modulating social novelty preference, a recent study demonstrated that corticotropin-releasing hormone released by infralimbic (IL) neurons projecting to the rostral LS acts to suppress investigation of a

familiar conspecific, thus promoting a relative increase in investigation of a novel conspecific (*de León Reyes et al., 2023*). Beyond social novelty/familiarity behaviors, the lateral septum plays a key role in modulating aggression (*Leroy et al., 2018*; *Wong et al., 2016*; *Guo et al., 2023*; *Albert and Chew, 1980*) and in modulating social reward seeking (*Li et al., 2023*). Dorsal LS neurons are known to inhibit ventral LS neurons via local collaterals (*Leroy et al., 2018*; *Wong et al., 2016*). Based on this evidence, it has been hypothesized that lateral inhibition between various LS populations could allow for selecting one behavioral output while suppressing the other (*Besnard and Leroy, 2022*; *Rizzi-Wise and Wang, 2021*). For example, increased activity in the ventral LS could suppress dorsal LS and disinhibit regions downstream of the dorsal LS, such as the NAc and the VTA thus promoting increased social approach and investigation while simultaneously suppressing aggression (*Leroy et al., 2018*). Thus, providing an exciting framework in which the LS could play a key role in regulating mutually exclusive social behaviors (*Besnard and Leroy, 2022*; *Rizzi-Wise and Wang, 2021*).

Although we focused largely on social novelty-seeking behavior, there is evidence that the hippocampal-septal circuitry plays a role in context-dependent modulation of behavior in a variety of contexts (*Besnard et al., 2019*; *Besnard et al., 2020*; *Décarie-Spain et al., 2022*). Although, we saw no effects in the ability of the mice to discriminate between novel and familiar foods with vHPC-LS inhibition in a discrimination assay in which the mice could smell but not access the food, a study found that activation of glutamatergic vHPC-LS neurons suppressed food intake in the home cage (*Sweeney and Yang, 2015*). Additionally, a recent study suggests that vHPC-LS neurons are necessary for mice to learn food reward-location associations (*Décarie-Spain et al., 2022*). It is unclear if this finding extends to socially rewarding stimuli. It also remains to be seen if the same vHPC-LS neurons/ subcircuits that are involved in mediating social preference also modulate other motivated behaviors.

In summary, we identify a novel vHPC-LS-VTA pathway that regulates social novelty preference. Our study provides insight into how social memory-related information is transformed into motivated social behaviors. Our study lays the groundwork for future experiments that could help us understand how these circuits might be disrupted in neurodegenerative disorders, like Alzheimer's disease, in which the ability of individuals to recognize and engage in appropriate social behaviors is dramatically affected.

## Methods

### Experimental model and subject details

All experimental procedures were conducted in accordance with the National Institutes of Health and Emory University's Institutional Animal Care and Use Committee (IACUC). All behavioral experiments were performed on male and female C57BL/6 J (Jax strain Number: 000664) mice aged ~4–12 weeks. Rabies tracing experiments included in *Figures 3 and 5* were performed on male mice with the animals in *Figure 5* belonging to the transgenic strains TH-Cre[+] (Jax strain Number: 008601). Mice were pair-housed or group-housed in cages with ad libitum access to water and chow (LabDiet). Mice were maintained on a 12 hr reverse light cycle. All experiments were conducted during their light-off period. In social discrimination tests, stimulus mice were sex and age-matched to the test animal.

### Stereotactic surgeries

Mice were placed in a stereotaxic setup (Kopf) and anesthetized with 1–2% isoflurane during surgery. All coordinates were sourced from *Paxinos and Franklin, 2019* and are described relative to bregma. Injections were performed using a Nanoject III (Drummond Scientific) and the virus was delivered at a rate of 2 nL per second. Virus injection coordinates are as follows, vHPC: 3 mm posterior, 3.25 mm lateral and –4.2 mm ventral relative to bregma, LS: 0.4 mm anterior, 0 mm lateral, –2.8 mm ventral relative to bregma, VTA: 3.1 mm posterior, 0.35 mm lateral, –4.7 mm ventral.

### DREADDs

For vHPC-LS chemogenetic silencing experiments, we bilaterally injected RetroAAV-hSyn-Cre (500 nL, Addgene Lot v70508, 3*10 *Besnard and Leroy, 2022*) into the LS. This was followed by 250 nl injections of either AAV5-hSyn-DIO-hM4D(Gi)-mCherry or AAV5-hSyn-DIO-mCherry into the vHPC of mice aged 5–6 weeks.

For LS-VTA chemogenetic silencing experiments, we bilaterally injected RetroAAV-hSyn-Cre (500 nL) into the VTA. This was followed by 250 nl injections of either AAV5-hSyn-DIO-mCherry or AAV5-hSyn-DIO-hM4D(Gi)-mCherry into the LS of mice aged 5–6 weeks.

## Optogenetics

For vHPC-LS cell body inhibition experiments, we bilaterally injected with RetroAAV-hSyn-Cre (500 nL, Addgene Lot v70508, 3*1013) into the LS. This was followed by 500 nL injections of either AAV5-Ef1-DIO-eNpHR3.0-EGFP (Addgene v32533, 1.1*10 13) or AAV5-hSyn-DIO-EGFP (Addgene, 1.1*1013) into the vHPC of mice aged 5–6 weeks. Optical fibers attached to ferrules were secured bilaterally using Metabond and dental acrylic to the skull to target the vHPC (Ferrule coordinates: –3.250 anterior, +/-3.0 lateral, –4.0 depth). Animals were allowed to recover and express the virus for 3 weeks prior to behavioral experiments.

## Retrograde tracing

To determine if LS neurons project to the VTA, we injected 750 nl of the retroAAV-Ef1a-mCherry-IRES-Cre virus into the VTA and 750 nl of the AAV5-CAG-FLEX-EGFP into the LS. Animals were allowed to recover and express the virus for 3 weeks prior to histology.

## Rabies tracing

For the LS-VTA monosynaptic rabies tracing experiments, 750 nl of the retroAAV-hsyn-Cre (500 nL, Addgene Lot v70508, 3×1013) was injected in the VTA of male and female C57BL/6 J mice. This was combined with the injection of 750 nl of the helper virus, AAV1.synP.FLEX.splitTVA.EGFP.B19G (Addgene, 2.4×1013) in the LS. Three weeks following injection, all mice were injected with 750 nl of the N2c-ΔG-deleted rabies virus ($5×10^8$, Thomas Jefferson University) (**Reardon et al., 2016**) expressing mCherry and pseudotyped with EnvA, RVdG-mCherry, into the LS.

For VTA$_{DA}$ monosynaptic rabies tracing experiments, 750 nl of the rabies helper virus, AAV1.synP.FLEX.splitTVA.EGFP.B19G (Addgene, 2.4×1013) was injected into the VTA of Th-Cre$^+$ mice (Jax strain number: 008601). Three weeks following injection, all mice were injected with 750 nl of the N2c-ΔG-deleted rabies virus expressing mCherry and pseudotyped with EnvA, RVdG-mCherry into the VTA ($\sim5 \times 10^8$, Thomas Jefferson University).

## Viral reagents

| ID | Lot # | Virus | Vendor | Titer (parts/ml) | Figure |
|---|---|---|---|---|---|
| 105553-AAVrg | v75884 | Retro AAV- hSyn-Cre | Addgene | 2.1×10^13 | 1, 3 F-K, 4 |
| 44362-AAV5 | v172066 | AAV5-hSyn-DIO-hM4D(Gi)-mCherry | Addgene | 2.4×10^13 | 1, 4 |
| 50459-AAV5 | v63478 | AAV5-hSyn-DIO-mCherry | Addgene | 8.4×10^12 | 1, 4 |
| 55632-AAVRg | v70508 | Retro AAV-Ef1a-mCherry-IRES.Cre | Addgene | 1.3×10^13 | 2, 3 A-C |
| 26966-AAV5 | v32533 | AAV5-Ef1a-DIO-eNpHR3.0-EYFP | Addgene | 1.1×10^13 | 2 |
| 51502-AAV5 | v60751 | AAV5-CAG-FLEX-EGFP | Addgene | 1.1×10^13 | 2, 3 A-C |
| 52473-AAV1 | v14715 | AAV1-synP-FLEX-TVA-EGFP-B19G | Addgene | 2.4×10^13 | 3 F-K, 5 |
| | SV-17–43 | Rabies-G-deleted-N2C-mCherry-EnvA | Thomas Jefferson University | 8×10^8 | 3 F-K, 5 |

## Histology and imaging

Mice were anesthetized with Euthasol (0.1 mg/kg) then transcardially perfused using PBS (0.5 X) followed by 4% PFA. Brains were extracted, left in PFA overnight, then transferred to a 30% sucrose solution the following day. A microtome (Leica Biosystems) was used to slice the coronal sections (50 um) and validate viral targeting. Slices were mounted using DAPI Fluoromount (Southern Biotech). All mounted sections were imaged using a high-throughput automated stitching fluorescent microscope (Keyence BZ-X810). Native fluorescence of expressed fluorophores was used to confirm expression.

For the chemogenetic silencing experiments, coronal sections of LS (LS-VTA experiments - *Figure 4*) and coronal sections of vHPC (vHPC-LS experiments - *Figure 1*) were used to confirm virus expression. Six mice were excluded from the vHPC-LS cohort for showing no expression or off-targeting. Seven mice were excluded from the LS-VTA cohort for showing no expression or off-targeting.

For cell body optogenetic inhibition experiments (*Figure 2*), coronal sections of vHPC were used to confirm virus expression, nuclear exclusion of NpHR, and to verify targeting of the fibers to vHPC (*Figure 2 - Figure Supplement 1*). Four mice were excluded from the vHPC-LS optogenetic cohort for showing no expression or off-targeting.

For monosynaptic rabies tracing experiments (*Figures 3 and 5*), brain sections ranging from the entire brain (~3.5 mm anterior bregma to ~4.5 mm posterior bregma) in ~100 um spacing were mounted. Sections were coverslipped with a DAPI Fluoromount mounting media and imaged using an automated widefield whole slide scanner (Keyence BZ810). Input neurons were mCherry labeled and starter cells expressed both EYFP (helper virus) and mCherry (rabies virus) (*Figures 3 and 5*).

## Behavioral assays

### Social discrimination task (SDT)

All mice used in this assay were ~8 weeks old. The social discrimination chamber (58.42 cm × 25.4 cm × 22.86 cm) contains two cages housing age and sex-matched stimulus mice (cage diameter = 7.62 cm) at opposite ends of the chamber. Prior to being run on the social discrimination assay, the test mouse was co-housed with a same-sex, age-matched conspecific for 72 hr to familiarize. The test mice were then allowed to explore the chamber containing both a novel and familiar stimulus mouse for 5 min. The location of the novel and familiar mice (left and versus right) was counterbalanced to eliminate side preferences. Behavior was tracked using an overhead camera and the position of the mouse in the arena was tracked using the open-source software Bonsai (*Lopes et al., 2015*). Percent time spent was calculated by measuring the amount of time the test mouse spent around each encaged social target (diameter of the social zone - 60 mm) relative to the entire assay duration.

For chemogenetic experiments (*Figures 1 and 4*), ~3–4 weeks after viral injection, mice expressing either the inhibitory DREADDs or mCherry received intraperitoneal injections of either Clozapine-N-Oxide (HelloBio, 1 mg/kg, i.p.) or saline (0.2 mL, i.p.) 30 min prior to being run on the behavioral assay. Mice were not given more than one CNO injection per day.

For vHPC-LS cell body optogenetic inhibition experiments (*Figure 2*), we used the Bonsai software to track the position of the test mouse in real-time and used this information to specifically inhibit the activity of vHPC-LS neurons only in the proximity of a particular conspecific (social zone diameter: 60 mm). The real-time tracking data from Bonsai was used to gate a green laser source via a TTL driver (Pulse Pal, Sanworks). Thus, the green laser source connected to the implanted fibers (532 nm, 6 mW, constant light - at the fiber tip) was turned on when the animal moved into only one of the social zones (either novel or familiar) and remained on until the animal moved out of the zone. The other social zone remained as a control zone and was not paired with light. Time spent by the mice in each zone was later estimated as described above.

For the novel versus familiar comparisons, both NpHR and EGFP mice were run on this assay across several conditions. Condition 1 - control condition- no stimulation paired with either the novel or the familiar mouse. Condition 2 - stimulation around the novel but not familiar animals. Condition 3 - stimulation around the familiar but not novel animal.

The same NpHR and EGFP mice were also run through a novel vs novel mouse experiment. In this experiment, the zone around one of the two novel mice was randomly designated as the stimulation zone, and entry into this zone was paired with green light illumination (light conditions same as above). Time spent by the mice in each of the two zones was estimated using Bonsai.

## Food discrimination task

Arena and handling procedures are identical to SDT. Familiar food was the chow (3 g, LabDiet), while the novel food was 3 g of fruit-flavored cereal. Food was placed under wired cages, such that mice could sniff but not taste the food. Mice were given 5 min to explore the arena containing the novel and familiar food. Percent time spent around the novel and familiar food was calculated by measuring the amount of time the test mouse spent around the cage containing the food (diameter of food zone - 6 cm) relative to the entire assay duration.

## Object discrimination task

Arena and handling procedures are identical to SDT. An object was placed into the home cage of test mice for 72 hr of familiarization. The objects were placed on the opposite ends of the discrimination chamber (without wired cages), and mice were given 5 min to explore the novel and familiar objects. For chemogenetic manipulation, procedures were performed identically to the SDT. Novel and familiar objects were placed on either the left or the right side of the chamber in a counterbalanced fashion.

For spatially-restricted optogenetic manipulation, mice were allowed to investigate two novel objects in an arena for 5 min. In the light ON condition, one of the two objects picked randomly was paired with green light stimulation (532 nm, 6 mW at the fiber tip constant light stimulation). Behavior recording conditions and laser conditions were identical to SDT.

## Open field

Mice were allowed to freely explore a large square chamber (46.64 cm × 46.64 cm) for 10 min. We determined the amount of time mice spent in the middle of the arena (11.14 cm × 11.14 cm) using Bonsai and MATLAB. For chemogenetic experiments, mice were injected with either saline or CNO 30 min prior to the test. For the vHPC-LS cell body inhibition experiment, the open field task was 9 min long. Mice performed the assay for 3 min with no light stimulation, followed by 3 min of green light stimulation (532 nm, 6 mW at the tip of the optical fiber; constant light illumination) and terminated with an additional 3 min of no light.

## Analyses

### Discrimination score

The discrimination score was calculated by the below equation. Time spent in proximity to novel minus ($N_{Time\ Spent}$) time spent in proximity to familiar ($F_{Time\ Spent}$), divided by the total duration of time spent in both zones ($Total_{Time\ Spent}$).

$$\frac{N_{time\ spent} - F_{time\ spent}}{Total_{time\ spent}}$$

Therefore, positive discrimination scores equate a mouse's preference for novel conspecifics and novel food odors and negative scores indicate an animal's preference for the familiar conspecific and food odors.

## Velocity

The velocity of animals within the Open Field task was determined by calculating distance traveled per unit of time and centroid positioning using Bonsai, an open-source tracking software (*Lopes et al., 2015*). Velocity was determined by change in pixels moved over change in seconds.

## Whole brain

The open-source WholeBrain software package in R allows for the annotation, analysis, and visualization of cellular resolution tracing in an interactive brain atlas (*Fürth et al., 2018*). The Whole Brain software was used specifically to register individual brain sections from animals onto the Allen common coordinate framework. It was used to map input neurons in rabies tracing experiments (*Figures 3 and 5*).

## Clozapine-N-Oxide

In the projection-specific inactivation experiments, all mice received CNO (Sigma; 1 mg/kg, i.p., in 2% DMSO and saline, 1 ml/ 100 g), regardless of virus condition, to equally expose animals to any unintended consequences of CNO. CNO was always administered 30 min prior to the behavioral experiment.

## Statistics

Analyses were performed using PRISM GraphPad, MATLAB, and RStudio. One-way ANOVA or unpaired t-tests were used to compare discrimination scores and velocity. Two-factor ANOVAs were used for social discrimination scores and open-field analyses. In the case of unequal variances,

Welch's ANOVA was used. Tukey's post hoc tests or paired t-tests were used in the case of significant interactions or main effects with >2 groups and are indicated in the figure with p-value annotations being *p<0.05, **p<0.01, ***p<0.001. Unless otherwise indicated, all tests are two-tailed. Please see *Supplementary file 1* for complete statistical analyses.

## Materials availability statement

No new materials were generated during the course of the study.

## Acknowledgements

We would like to thank Dr. Annabelle Singer and Dr. Shannon Gourley for feedback on this manuscript. This work was supported by National Institutes of Health grants R01MH130755 (MM) and F31MH133373 (JI).

## Additional information

### Funding

| Funder | Grant reference number | Author |
|---|---|---|
| National Institutes of Health | R01MH130755 | Malavika Murugan |
| National Institutes of Health | F31MH133373 | Jennifer Isaac |

The funders had no role in study design, data collection and interpretation, or the decision to submit the work for publication.

### Author contributions

Maha Rashid, Conceptualization, Data curation, Software, Formal analysis, Validation, Investigation, Visualization, Methodology, Writing – original draft, Writing – review and editing; Sarah Thomas, Data curation, Formal analysis, Validation, Investigation, Methodology; Jennifer Isaac, Data curation, Software, Formal analysis, Supervision, Validation, Investigation, Methodology; Sonia Corbett Karkare, Data curation, Software, Formal analysis, Validation, Investigation, Methodology; Hannah Klein, Investigation; Malavika Murugan, Conceptualization, Supervision, Funding acquisition, Writing – original draft, Project administration, Writing – review and editing

### Author ORCIDs

Sonia Corbett Karkare ⓘ https://orcid.org/0000-0002-2590-4597
Malavika Murugan ⓘ https://orcid.org/0000-0003-1300-0379

### Ethics

All experimental procedures were approved by the Emory Institutional Animal Care and Use Committee (PROTO202000014).

Reviewer #1 (Public Review): https://doi.org/10.7554/eLife.97259.2.sa1
Reviewer #2 (Public Review): https://doi.org/10.7554/eLife.97259.2.sa2
Author response https://doi.org/10.7554/eLife.97259.2.sa3

## Additional files

### Supplementary files

Supplementary file 1. Detailed statistical reporting for *Figures 1–5*.

MDAR checklist

Data availability

The data shown in the figures are available as source data files.

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
