## [Editor Report · eLife assessment]

This **important** manuscript uses circuit mapping, chemogenetics, and optogenetics to demonstrate a novel hippocampal lateral septal circuit that regulates social novelty behaviours and shows that downstream of the hippocampal septal circuit, septal projections to the ventral tegmental area are necessary for general novelty discrimination. The strength of the evidence supporting the claims is **convincing** but would be strengthened by the inclusion of additional functional assays. The work will be of interest to systems and behavioural neuroscientists who are interested in the brain mechanisms of social behaviours.

---

## [Referee Report · Reviewer #1 (Public Review)]

Summary:

The study investigated the neural circuits underlying social novelty preference in mice. Using viral circuit tracing, chemogenetics, and optogenetics in the vHPC, LS, and VTA, the authors found that vHPC to LS projections may contribute to the salience of social novelty investigations. In addition, the authors identify LS projections to the VTA involved in social novelty and familiar food responses. Finally, via viral tracing, they demonstrate that vHPC-LS neurons may establish direct monosynaptic connections with VTA dopaminergic neurons. The experiments are well-designed, and the conclusions are mostly very clear. The manuscript is well-written and logically organized, and the content will be of interest to specialists in the field and to the broad readership of the journal.

Strengths:

(1) The vHPC has been involved in social memory for novel and familiar conspecifics. Yet, how the vHPC conveys this information to drive motivation for novel social investigations remains unclear. The authors identified a pathway from the vHPC to the LS and eventually the VTA, that may be involved in this process.

(2) Mice became familiar with a novel conspecific by co-housing for 72h. This represents a familiarization session with a longer duration as compared to previous literature. Using this new protocol, the authors found robust social novelty preference when animals were given a choice between a novel and familiar conspecific.

(3) The effects of vHPC-LS inhibition are specific to novel social stimuli. The authors included novel food and novel object control experiments and those were not affected by neuronal manipulations.

(4) For optogenetic studies, the authors applied closed-loop photoinhibition only when the animals investigated either the novel conspecific or the familiar. This optogenetic approach allowed for the investigation of functional manipulations to selective novel or familiar stimuli approaches.

Weaknesses:

(1) The abstract and the overall manuscript pose that the authors identified a novel vHPC-LS-VTA pathway that is necessary for mice to preferentially investigate novel conspecifics. However, the authors assessed the functional manipulations of vHPC-LS and LS-VTA circuits independently and the sentence could be misleading. Therefore, a viral strategy specifically designed to target the vHPC-LS-VTA circuit combined with optogenetic/chemogenetic tools and behavior may be necessary for the statement of this conclusion.

(2) The authors combined males and females in their analysis, as neural circuit manipulation affected novelty discrimination ratios in both sexes. However, supplementary Figure 1 demonstrates the chemogentic inhibition of vHPC-LS circuit may cause stronger effects in male mice as compared to females.

(3) In most experiments, the same animals were used for social novelty preference, for food or object novelty responses but washout periods between experiments are not mentioned in the methods section. In this line, the authors did not mention the time frame between the closed-loop optogenetic experiments that silenced the vHPC-LS only during familiar and then only novel social investigations. When using the same animals tested for social experiments in the same context there may be an effect of context-dependent social behaviors that could affect future outcomes.

(4) All the experiments were performed in a non-cell-type-specific manner. The viral strategies used targeted multiple neuronal subpopulations that could have divergent effects on social novelty preference. This constraint could be added in the discussion section.

(5) The authors' assumptions were all based on experiments of necessity. The authors could use an experiment of sufficiency by targeting for instance the LS-VTA circuit and assess if animals reduce novel social investigations with LS-VTA photostimulation.

---

## [Referee Report · Reviewer #2 (Public Review)]

Summary:

Rashid and colleagues demonstrate a novel hippocampal lateral septal circuit that is important for social recognition and drives the exploration of novel conspecifics. Their study spans from neural tracing to close-loop optogenetic experiments with clever controls and conditions to provide compelling evidence for their conclusion. They demonstrate that downstream of the hippocampal septal circuit, septal projections to the ventral tegmental area are necessary for general novelty discrimination. The study opens an avenue to study these circuits further to uncover the plasticity and synaptic mechanisms regulating social novelty preference.

Strengths:

Chemogenetic and optogenetic experiments have excellent behavioral controls. The synaptic tracing provides important information that informs the narrative of experiments presented and invites future studies to investigate the effects of septal input on dopaminergic activity.

Weaknesses:

There are unclear methodological important details for circuit manipulation experiments and analyses where multiple measures are needed but missing. Based on the legends, the chemogenetic experiment is done in a within-animal design. That is the same mouse receives SAL and CNO. However, the data is not presented in a within-animal manner such that we can distinguish if the behavior of the same animal changes with drug treatment. Similarly, the methods specify that the optogenetic manipulations were done in three different conditions, but the analyses do not report within-animal changes across conditions nor account for multiple measures within subjects. Finally, it is unclear if the order of drug treatment and conditions were counterbalanced across subjects.

---

## [Author Response]

**eLife Assessment**
This important manuscript uses circuit mapping, chemogenetics, and optogenetics to demonstrate a novel hippocampal lateral septal circuit that regulates social novelty behaviours and shows that downstream of the hippocampal septal circuit, septal projections to the ventral tegmental area are necessary for general novelty discrimination. The strength of the evidence supporting the claims is convincing but would be strengthened by the inclusion of additional functional assays. The work will be of interest to systems and behavioural neuroscientists who are interested in the brain mechanisms of social behaviours.

We thank the reviewers for their thoughtful and constructive feedback. We are excited that both reviewers thought that the manuscript was of “interest to specialists in the field and to the broad readership of the journal”, that the paper was “well-written and logically organized” and that the “study opens an avenue to study these circuits further to uncover the plasticity and synaptic mechanisms regulating social novelty preference.” Additionally, the reviewers wrote that the experiments were “well-designed” “with clever controls and conditions to provide compelling evidence for their conclusion.” The reviewers additionally provided constructive feedback, which we address in our responses below.

**Public Reviews:**

**Reviewer #1 (Public Review):**
Summary:The study investigated the neural circuits underlying social novelty preference in mice. Using viral circuit tracing, chemogenetics, and optogenetics in the vHPC, LS, and VTA, the authors found that vHPC to LS projections may contribute to the salience of social novelty investigations. In addition, the authors identify LS projections to the VTA involved in social novelty and familiar food responses. Finally, via viral tracing, they demonstrate that vHPC-LS neurons may establish direct monosynaptic connections with VTA dopaminergic neurons. The experiments are well-designed, and the conclusions are mostly very clear. The manuscript is well-written and logically organized, and the content will be of interest to specialists in the field and to the broad readership of the journal.Strengths:(1) The vHPC has been involved in social memory for novel and familiar conspecifics. Yet, how the vHPC conveys this information to drive motivation for novel social investigations remains unclear. The authors identified a pathway from the vHPC to the LS and eventually the VTA, that may be involved in this process.(2) Mice became familiar with a novel conspecific by co-housing for 72h. This represents a familiarization session with a longer duration as compared to previous literature. Using this new protocol, the authors found robust social novelty preference when animals were given achoice between a novel and familiar conspecific.(3) The effects of vHPC-LS inhibition are specific to novel social stimuli. The authors included novel food and novel object control experiments and those were not affected by neuronal manipulations.(4) For optogenetic studies, the authors applied closed-loop photoinhibition only when the animals investigated either the novel conspecific or the familiar. This optogenetic approach allowed for the investigation of functional manipulations to selective novel or familiar stimuli approaches.Weaknesses:(1) The abstract and the overall manuscript pose that the authors identified a novel vHPC-LS-VTA pathway that is necessary for mice to preferentially investigate novel conspecifics. However, the authors assessed the functional manipulations of vHPC-LS and LS-VTA circuits independently and the sentence could be misleading. Therefore, a viral strategy specifically designed to target the vHPC-LS-VTA circuit combined with optogenetic/chemogenetic tools and behavior may be necessary for the statement of this conclusion.

The reviewer raises an important point. Although Figure 3 shows that vHPC (vCA1 and vCA3) is the source of the greatest number of monosynaptic inputs onto LS-VTA neurons, we did not perform any experiments that specifically manipulated vHPC neurons that project to LS-VTA neurons. While these experiments would be extremely interesting, they are technically challenging and beyond the scope of this study.

(2) The authors combined males and females in their analysis, as neural circuit manipulation affected novelty discrimination ratios in both sexes. However, supplementary Figure 1 demonstrates the chemogentic inhibition of vHPC-LS circuit may cause stronger effects in male mice as compared to females.

The reviewer makes an interesting point. We can confirm that we found no significant differences in the effectiveness of our vHPC-LS inhibition between the males and females (2-factor ANOVA with sex (male/female) and drug condition (saline/CNO) as factors on the discrimination scores of hM4Di expressing animals: interaction p=0.2241, sex: p=0.1233, drug condition: p=0.0166). These data suggest that there are no significant sex differences in the effectiveness of inhibition of the vHPC-LS neurons.

(3) In most experiments, the same animals were used for social novelty preference, for food or object novelty responses but washout periods between experiments are not mentioned in the methods section. In this line, the authors did not mention the time frame between the closed-loop optogenetic experiments that silenced the vHPC-LS only during familiar and then only novel social investigations. When using the same animals tested for social experiments in the same context there may be an effect of context-dependent social behaviors that could affect future outcomes.

We thank the reviewer for this important clarification. We apologize for not including these crucial details in our Methods section. For both the chemogenetic and optogenetic inhibition experiments, all conditions were separated by a minimum of 24 hours. In the chemogenetic inhibition experiments, saline and CNO conditions were counterbalanced between animals. Similarly, we counterbalanced the order of light ON vs light OFF conditions across animals during our optogenetic inhibition experiments.

(4) All the experiments were performed in a non-cell-type-specific manner. The viral strategies used targeted multiple neuronal subpopulations that could have divergent effects on social novelty preference. This constraint could be added in the discussion section.

The reviewer raises an important point. In our study, while we specifically manipulate projection populations (either vHPC-LS or LS-VTA), it is possible that these projection populations themselves are composed of heterogeneous cell types. It would be an interesting direction of study to pursue in the future.

(5) The authors' assumptions were all based on experiments of necessity. The authors could use an experiment of sufficiency by targeting for instance the LS-VTA circuit and assess if animals reduce novel social investigations with LS-VTA photostimulation.

We agree with the reviewers that it would be interesting to determine if LS-VTA neurons are sufficient, in addition to being necessary, to drive social novelty. These will be interesting experiments to pursue in the future.

**Reviewer #2 (Public Review):**
Summary:Rashid and colleagues demonstrate a novel hippocampal lateral septal circuit that is important for social recognition and drives the exploration of novel conspecifics. Their study spans from neural tracing to close-loop optogenetic experiments with clever controls and conditions to provide compelling evidence for their conclusion. They demonstrate that downstream of the hippocampal septal circuit, septal projections to the ventral tegmental area are necessary for general novelty discrimination. The study opens an avenue to study these circuits further to uncover the plasticity and synaptic mechanisms regulating social novelty preference.Strengths:Chemogenetic and optogenetic experiments have excellent behavioral controls. The synaptic tracing provides important information that informs the narrative of experiments presented and invites future studies to investigate the effects of septal input on dopaminergic activity.Weaknesses:There are unclear methodological important details for circuit manipulation experiments and analyses where multiple measures are needed but missing. Based on the legends, the chemogenetic experiment is done in a within-animal design. That is the same mouse receives SAL and CNO. However, the data is not presented in a within-animal manner such that we can distinguish if the behavior of the same animal changes with drug treatment. Similarly, the methods specify that the optogenetic manipulations were done in three different conditions, but the analyses do not report within-animal changes across conditions nor account for multiple measures within subjects.

Thank you for raising this important point. We agree that a repeated measures ANOVA would be ideal, but there is sufficient behavioral variability that such analyses will be difficult without very large sample sizes.

Finally, it is unclear if the order of drug treatment and conditions were counterbalanced across subjects.

As mentioned in the above response to Reviewer 1, for both the chemogenetic and optogenetic inhibition experiments, all conditions were separated by a minimum of 24 hours and we counterbalanced the order of chemogenetic (saline/CNO) and optogenetic (light ON/light OFF) experimental manipulations across animals.